# Comparative Analysis for Genetic Characterization in Korean Native Jeju Horse

**DOI:** 10.3390/ani11071924

**Published:** 2021-06-28

**Authors:** Wooseok Lee, Seyoung Mun, Song-Yi Choi, Dong-Yep Oh, Yong-Soo Park, Kyudong Han

**Affiliations:** 1Center for Bio-Medical Engineering Core Facility, Dankook University, Cheonan 31116, Korea; wooseoklee87@gmail.com (W.L.); munseyoung@gmail.com (S.M.); 2Department of Nanobiomedical Science & BK21 PLUS NBM Global Research Center for Regenerative Medicine, Dankook University, Cheonan 31116, Korea; 3Department of Pathology, Colleage of Medicine, Chungnam National University, Daejeon 34134, Korea; sychoi@cnu.ac.kr; 4Livestock Research Institute, Gyeongsangbuk-Do, Yeongju 36052, Korea; ody1234@korea.kr; 5Department of Equine Industry, Korea National College of Agriculture and Fisheries, Jeonju 54874, Korea; 6Department of Microbiology, College of Science and Technology, Dankook University, Cheonan 31116, Korea

**Keywords:** Jeju horse, Thoroughbred, WGRS, SNP, eqCD1a6, digital PCR, *Equus ferus caballus*

## Abstract

**Simple Summary:**

In modern times, horse breeds, mostly in horse racing, are the Thoroughbred varieties obtained by breeding three Godolphin Arabians with British mares in England. Especially in Jeju Island, Korea, Jeju horses have been introduced from Mongolia since the 13th century. They have contributed a lot to the agricultural community, but their population has been rapidly decreasing due to rapid agricultural industrialization. Therefore, we sympathize with Jeju horse-specific genetic variation and compare and analyze evolutionary correlations by utilizing Whole Genome Sequencing analysis to evaluate the genetic diversity of Jeju horses and preserve genetic information. We explored Jeju horse-specific genetic differences through a comparative analysis of large-capacity genomic data between the public database and a Thoroughbred variety. In order to adapt to the barren external environment, it is predicted that Jeju horses have experienced strong positive selection in the direction of accumulating many genetic variations, enough to cause functional differences in the eqCD1a6 gene to have an efficient immune function. In addition, we further validate the Jeju horse-specific single nucleotide polymorphisms in the aqCD1a6 gene by employing the digital PCR method, a diagnostic technique for genetic variations.

**Abstract:**

The Jeju horse is a native Korean species that has been breeding on Jeju Island since the 13th century. Their shape has a distinct appearance from the representative species, Thoroughbred. Here, we performed a comparison of the Jeju horse and Thoroughbred horse for the identification of genome-wide structure variation by using the next-generation sequencing (NGS) technique. We generated an average of 95.59 Gb of the DNA sequence, resulting in an average of 33.74 X sequence coverage from five Jeju horses. In addition, reads obtained from WGRS data almost covered the horse reference genome (mapped reads 98.4%). Based on our results, we identified 1,244,064 single nucleotide polymorphisms (SNPs), 113,498 genomic insertions, and 114,751 deletions through bioinformatics analysis. Interestingly, the results of the WGRS comparison indicated that the eqCD1a6 gene contains signatures of positive natural selection in Jeju horses. The eqCD1a6 gene is known to be involved in immunity. The eqCD1a6 gene of Jeju horses commonly contained 296 variants (275 SNPs and 21 INDELs) that were compared with its counterpart of two Thoroughbred horses. In addition, we used LOAA, digital PCR, to confirm the possibility of developing a molecular marker for species identification using variant sites. As a result, it was possible to confirm the result of the molecular marker with high accuracy. Nevertheless, eqCD1a6 was shown to be functionally intact. Taken together, we have found significant genomic variation in these two different horse species.

## 1. Introduction

The emergence of the family *Equidae* has a longer evolutionary history than the emergence of *Homo sapiens*. The evolution of horses occurred over ~50 million years. At first, people considered horses to be hunting targets, but those who found out their abilities began to domesticate them [1,2,3]. The horse was domesticated and used for farming, transportation, food, and war purposes. After the Middle Ages, domestic horses were widely distributed and were commonly used for human hobbies such as riding and racing. As a result, the Thoroughbred horse has become widespread as the most representative horse today through pedigree management. The Thoroughbred horse has continued to manage superior objects in the same way as single nucleotide polymorphism (SNP) and restriction fragment length polymorphism (RFLP) [4]. Recently, several types of research using next-generation sequencing (NGS) technology, such as whole-genome sequencing, target sequencing, and RNA sequencing, to characterize genetic features of different horses have been conducted [5,6,7,8,9]. However, genetic research on Jeju horses using NGS has been relatively less studied.

Jeju horse is a general term for the ponies that grow wild on Jeju Island. It was designated as National Monument number 347 on 8 February 1986, in Korea. It is estimated to be a hybrid of the pony that used to live on Jeju Island and the Mongolian horse that flowed into the island from the late 1200s to the 1300s [7,10]. The average body height of an adult Jeju horse over five years old is 117 cm for females and 115.3 cm for males, which is smaller than a mixed horse or modern horse (crossbreed or improved breed), and differs genetically from other horses [11]. The fur of the Jeju horse is brown, reddish-brown, gray, black, light yellow, and stained, with the brown individuals being the most common, followed by reddish-brown. In addition, the Jeju horse has a low front, high backside, medium body, and a large chest ratio, which is a typical body shape suitable for a sumpter (Figure 1).

Jeju horses have been close to extinction in the past. However, the protection afforded by the Endangered Species Act means that the population of Jeju horses has rebounded today, with more than 5000 Jeju horses breeding [12]. Most studies on Jeju horses are morphological compared to Thoroughbred horses. Furthermore, the most basic genetic studies for the academic establishment of Jeju horses are far from insufficient [13].

In this study, we performed whole-genome resequencing (WGRS) of five domesticated Jeju horses and one Thoroughbred. Using the horse reference genome data, we determined a significant number of SNPs and insertion/deletion (INDEL) throughout the genome. All the structure variants were annotated, particularly resulting in nonsynonymous mutations that can be further used as genetic markers that would predict phenotypic variation in patterns of interest in Jeju horses. Interestingly, the results of the WGRS comparison indicated that the eqCD1a6 gene contains signatures of positive natural selection in Jeju horses and is thought to be the result of environmental adaptation. In addition, we confirmed the Gene Ontology (GO) term of Jeju horses through GO analysis and found that there are many correlations in genes related to the heart and nerves. These results are indirect evidence that Jeju horses show physical differences from Thoroughbred horses.

## 2. Materials and Methods

### 2.1. Sample Preparation and DNA Extraction

Genomic DNAs (gDNAs) were extracted from whole-blood samples (each 200 uL) of five Korean native horses (Jeju horses) and one Thoroughbred using a GenEx Blood kit (GeneAll, Seoul, Korea) according to the manufacturer’s guidelines. The blood samples were provided by the Korea National College of Agriculture and Fisheries (Jeonju, Korea). All experimental procedures were reviewed and approved by animal experimental ethics committee of Kyungnam National University of Science and Technology (Jinju-Si, Republic of Korea). The gDNA qualities of the six horses were evaluated using a microvolume spectrometer Colibri (Titertek-Berthold, Pforzheim, Germany) at the Center for Bio-medical Engineering Core Facility (Dankook University, Cheonan, Korea). We used all samples only with high-quality DNA (260/230 ratio ≥ 1.8, 260/280 ratio ≥ 1.8) for sequencing.

### 2.2. Whole-Genome Resequencing (WGRS)

DNA libraries for WGRS were constructed using the TruSeq Nano DNA Library Prep kit (Illumina, San Diego, CA, USA). Before constructing the library, to determine the concentration of double-stranded DNA (dsDNA), a Qubit 4 Fluorometer (Thermo Fisher, Waltham, MA, USA) was used to measure and dilute the samples accurately. Each dsDNA sample (100 ng) was randomly sheared by a Qsonica Q800R3 Ultrasonication System (Qsonica, Newtown, CT, USA) and was verified by the Agilent Bioanalyzer 2100 (Agilent Technologies, Santa Clara, CA, USA) with a High Sensitivity DNA chip in size range of 300 to 400 bp according to manufacturer’s protocol. The sheared DNAs were subjected to end-repair, 3′ adenylation, and Illumina’s Universal adaptor ligation. Finally, the libraries were amplified using the Illumina Universal primer cocktail. We confirmed the library quality check using the Agilent Bioanalyzer 2100 (Agilent Technologies, Santa Clara, CA, USA) with a High Sensitivity DNA chip. Each DNA sequencing library was sequenced by an Illumina HiSeq 2500 platform (Illumina, San Diego, CA, USA). All processes were conducted by Theragen Bio Institute (Suwon, Korea).

### 2.3. Data Analysis

Raw reads in FASTQ format generated by WGRS were filtered with a cut-off of Q20 using FastQC and Sickle (https://www.bioinformatics.babraham.ac.uk/projects/fastqc and https://github.com/najoshi/sickle, respectively accessed on 8 January 2018) trimmed adapter sequences from sequencing reads. To align sequencing reads, these were mapped to the horse reference genome assembly (September 2007, Broad/equCab2) using Burrow Wheeler Aligner (BWA) [14] with the standard options. The reference genome data were downloaded from the Ensembl site (www.ensembl.org accessed on 7 January 2018). After that, we used the Picard tool (MarkDuplicates and FixMateInformation) to remove the PCR duplicated reads, and as a result, paired-end mismatches were corrected. After the sequenced genomes were mapped to the reference genome, we used the GATK tool [15] to re-calibrate the output, which improved the accuracy of the variants. As a correction, we carried out RealignerTargetCreator, and this performance was to realign the reads by reanalyzing the callings that are supposed to be variants. The presence of variants was frequently generated by mapping errors so that further corrections could yield more robust variants. Using the GATK tool, SNPs and INDELs, which are sequence variations for each sample, were identified. For each locus of variants identified through GATK, SnpEff analysis was performed to annotate genetic variations. Finally, the variant data (SNPs and INDELs) of each sample were compared using awk and Python, the scripting languages of Linux. Thus, Jeju horse-specific SNPs and INDELs were analyzed.

### 2.4. PCR and DNA Sequencing Analysis

In order to confirm the newly discovered Jeju horse-specific eqCD1a6 gene in this study, we performed PCR amplification using gDNAs from 17 Thoroughbred DNAs, 15 Jeju horse DNAs, and 3 Halla horse DNAs. The horse gDNA samples were provided by the Korea National College of Agriculture and Fisheries. Oligonucleotide primers for each PCR reaction were designed by using the software Primer3Plus (http://www.bioinformatics.nl/cgi-bin/primer3plus/primer3plus.cgi accessed on 05 March 2019) and Oligocalc (http://biotools.nubic.northwestern.edu/OligoCalc.html accessed on 05 March 2019) program [16]. PCR amplification of each Jeju horse-specific eqCD1a6 locus was performed in a 20 uL reaction using 10 ng gDNA, 10 pM of oligonucleotides primers, and 10 uL of the master mixture of 2 × Lamp Taq PCR Master Mix 4 (BioFACT™, Daejeon, Korea). The PCR amplification was started with an initiation of 5 min at 95 °C, followed by 35 cycles of 30 s at 95 °C, 40 s at the optimal annealing temperature and optimal time depending on PCR product size for extension at 72 °C depending on each product size, and final extension of 5 min at 72 °C (Bio-Rad laboratories, Berkeley, CA, USA) at the Center for Biomedical Engineering Core Facility (Dankook University, Korea). The PCR products (5 uL) were confirmed through electrophoresis with 1% agarose gel for electrophoresis, stained with EcoDye Nucleic acid staining solution (BioFACT, Daejeon, Korea), and visualized with UV fluorescence. The information of primer pairs is described in Appendix A.

For each sample, Sanger sequencing was performed to validate the Jeju horse-specific eqCD1a6 gene sequenced. DNA sequencing was performed on an Applied Biosystems AB3500XL Genetic Analyzer (Applied Biosystems, Waltham, MA, USA) at the Center for Biomedical Engineering Core Facility (Dankook University, Korea). The Sanger sequencing data were analyzed using Bioedit (version 7.2) to compare with the reference genome data.

### 2.5. Sliding-Window Analysis of dN/dS (Positive Selection Analysis)

Jeju horse and Thoroughbred eqCD1a6 genes were aligned using Bioedit. To calculate sliding-window dN/dS values for the alignment, the SLIDERKK program was used [17]. The sequence alignment of Jeju horse and Thoroughbred was analyzed with a window size of 135 bp and a sliding increment of 9 bp.

### 2.6. Genotyping Assay Using Digital PCR

To distinguish the Jeju horse more accurately, we applied the eqCD1a6 gene to a digital PCR platform (LOAA, Optolane, Korea). The FAM probe (SFCprobes, Yongin-si, Korea) was used for the detection of the Jeju horse genome. The FRET SFC620 probe (SFCprobes, Korea) was used for the detection of Thoroughbred genomes. The eqCD1a6 primer set (IDT, Coralville, IA, USA) was designed at the flacking sequence region of the FAM and FRET SFC620 probes (Appendix A).

The digital PCR reaction mixture (30 uL) contained 10 uL primer–probe mixture (Appendix A), 10 uL of 3X Dr. PCR premix, 6 ng DNA, and nuclease-free water up to 30 uL. The reaction mixtures were loaded into wells of LOAA Dr. Digital PCR cartridge (Optolane, Seongnam-si, Korea). Then, the cartridges were placed into the POSTMAN equipment (Optolane, Seongnam-si, Korea) for uniform application. Finally, we mounted the prepared cartridge on the LOAA equipment (Optolane, Seongnam-si, Korea), and we launched the thermocycling program. The digital PCR uses a 2-step cycling protocol, with these conditions: Uracil-DNA Glycosylase step of 3 min at 50 °C, initial denaturation step of 15 min at 95 °C, followed by 40 cycles of 95 °C for 10 s and 60 °C for 15 s. A total of 16,800 to 19,200 valid wells were created from each sample. We analyzed digital PCR results using the “Optolane OnPoint Pro” software (Optolane, Seongnam-si, Korea).

## 3. Results

### 3.1. Sequencing, Read Mapping, and Genomic Variant Detection

To detect Jeju horse-specific genomic variation, we compared the genomes of Jeju horses, which have been geographically isolated for a long time, with the horse reference genome. In order to obtain more accurate results, we generated additional genome data of a Thoroughbred horse breed in the same species as the horse reference genome. We performed WGRS on five Jeju horses and one Thoroughbred horse using the Illumina HiSeq 2500 platform and obtained a total of 696,089,910 raw reads. The raw reads were trimmed and deduplicated via Sickle and Picard, resulting in an average of 609,556,376 clean reads. Next, the clean reads were mapped to the horse reference genome using BWA, and finally, the average read depth of the six horses ranged from 31.48x to 46.01x (an average of ~35.79x, Table 1).

To identify Jeju horse-specific structural variation (SV), we compared the genome data obtained from five Jeju horses with the horse reference genome using the GATK tool and the variant annotations in VCF files were created with SnpEff. As a result, we found an average of 6,686,678 SNPs, 436,460 insertions, and 456,249 deletions in the Jeju horse (Table 2).

Next, we analyzed the SV positions shared by five Jeju horses to find Jeju horse-specific SVs. As a result, 2,758,242 SNPs, 233,819 insertions, and 240,738 deletions were identified as SVs shared by five Jeju horses (Figure 2). The SV results shared by five Jeju horses were compared with the WGRS data of one Thoroughbred horse to further reduce the number of Jeju horse-specific SVs by obtaining 1,244,064 SNPs, 113,498 insertions, and 114,751 deletions (Figure 3).

Furthermore, Jeju horse-specific SNPs were compared with SNP data from open-access databases (dbSNP, Ensembl, and Broad Institute) [18]. The previously registered SNPs consisted of 21,443,129 dbSNPs, 5,008,750 Ensembl, and 1,163,466 Broad Institute. Finally, a total of 408,601 Jeju horse-specific SNPs that do not overlap with open access databases were identified (Figure 4). The 408,601 Jeju horse-specific SNPs were divided into 94,192 homozygous and 314,409 heterozygous. For the 113,498 Jeju horse-specific insertions, they were divided into 85,394 homozygous and 28,104 heterozygous. For the 114,751 Jeju horse-specific deletions, they were divided into 75,115 homozygous and 39,636 heterozygous. Among the identified SVs, we analyzed the loci of those corresponding to homozygous SVs (94,192 homozygous SNPs, 85,394 homozygous insertions, 75,115 homozygous deletions) in all five Jeju horses. This showed that most SVs resided in intergenic regions (an average of 64%), followed by an intron, upstream region, and exon (Table 3).

### 3.2. Functional Annotation of Nonsynonymous

Among the numerous Jeju horse-specific SVs, we focused on nonsynonymous SNPs (724) and INDELs (564 insertions and 879 deletions) present in genic regions that may have a more functional impact. We performed GO term analysis using the ClueGO plugin of Cytoscape software [19] on 788 genes containing 2167 SVs (nonsynonymous SNPs and INDELs). As a result, 106 out of 788 genes were correlated with 13 GO categories (Table 4).

The 13 GO categories were mainly involved in cardiac development/blood circulation and neurodevelopment/hormone secretion. The reason why these two groups are characterized is probably due to the external differences between the Jeju horse and Thoroughbred horse. Thoroughbred horses have been gradually improved to be faster by humans. As a result, they have a body suitable for high speed, but their endurance was weakened. In contrast, Jeju horses are well known for their endurance. Due to these differences, we hypothesize that Jeju horses and Thoroughbred horses show significant differences in cardiac development/blood circulation and neurodevelopment/hormone secretion (Figure 5). Interestingly, we found that 275 SNPs and 21 INDELs common in five Jeju horses exist in the eqCD1a6 gene (Table 5).

The function of the CD1 gene family is unknown, but recently it has been known to be involved in immunity to Rhodococcus equi. R. equi is associated with Mycobacterium tuberculosis and is structurally similar to the nocardioform actinomycete bacterium [20]. M. tuberculosis is known as a cause of pulmonary tuberculosis in humans. In contrast, in young horses, R. equi is known to be a life-threatening pathogen that causes pyogranulomatous pneumonia [21]. Previous studies have reported that most mammals have more than one isoform of the CD1 gene [22]. In a previous study, 13 complete eqCD1 genes were identified in the horse genome, and they were largely divided into eqCD1a, eqCD1b, eqCD1c, eqCD1d, and eqCD1e. Among them, eqCD1a is the largest isoform group (eqCD1a1~eqCD1a7) [23]. The eqCD1a6 gene is 2281 bp in length and consists of six exons that encode the signal peptide, α-1, α-2, α-3, the transmembrane region, and the cytoplasmic tail. Of the 275 SNPs identified in Jeju horses, 51 SNPs were located in the exon regions of eqCD1a6 and the most SNPs were found at Exon 2 and Exon 3 (18 and 25 SNPs, respectively) (Figure 6). We compared the amino acid sequence of the eqCD1a6 of Jeju horses with its counterpart in the horse reference genome. As a result, the eqCD1a6 gene has accumulated 37 nonsynonymous changes, but no stop codons have been found. Most nonsynonymous changes occurred in α-1, α-2, and α-3 (14, 16, and 5 nonsynonymous substitutions, respectively) (Table 6).

### 3.3. Positive Selection of eqCD1a6 Gene in Jeju Horses

In order to confirm the correct Jeju horse-specific SNP of the eqCD1a6 gene, we used additional DNA samples of 35 horses (Jeju horse 15, Halla horse 3, and Thoroughbred (raised in Jeju Island) 17), which were used to PCR the exon region of eqCD1a6 and confirmed by Sanger sequencing. As a result, out of 51 SNPs, we identified 36 SNPs unique to Jeju horses, which most Jeju horse samples have in common in the Jeju horse genome. We conducted a dN/dS ratio analysis based on Jeju horse-specific SNP data [24] (Figure 7). The dN/dS ratios estimate the evolution rate by considering synonymous and nonsynonymous variations. The eqCD1a6 gene in Jeju horses appears to have been a positive selection to escape from the current stage when compared to Thoroughbred horses.

### 3.4. Genotyping Assay for Molecular Markers

Korea is conducting pedigree management to protect species and manage individual Jeju horses, and species identification is performed through the appearance of Jeju horses and various genetic tests. However, sometimes inaccurate results are obtained because Jeju horse genome data are not compared with other Jeju horses. We proceeded with the development of a molecular marker. Based on the Jeju horse-specific CD1a6 obtained through the research results, we expected that the development of a species-specific molecular marker would enable a more accurate and faster test, and digital PCR was applied to this. Digital PCR has become accessible and easier to handle due to recent advances in technology and the diversification of equipment. The LOAA equipment used in this study is a semiconductor method that is different from the existing droplet digital PCR and has ultra-fast, ultra-light, ultra-compact, and ultra-sensitive features. We designed the probe and primer set based on the SNPs of the eqCD1a6 gene with the most variation between the Jeju horse and Thoroughbred. For the sample, 10 gDNA samples were used for each Jeju horse and Thoroughbred. As a result, it was shown that the unique aspect of the Jeju horse was confirmed through the molecular marker, and the accuracy reached 80% (Figure 8 and Table 7).

In this study, since only one probe was used, it showed an accuracy of 80%. However, we predict that designing Jeju horse-specific probes based on sequence information of various SNPs will be a more accurate species molecular marker. Based on these results, it is thought that this experimental method can be applied to various fields. Furthermore, if digital PCR is as easy to operate and lightweight as LOAA, it is expected that digital PCR is expected to be fully utilized in point of care testing (PoCT).

## 4. Discussion

Jeju horses have been isolated from the island environment and bred with little contact with other species of horses. As a result, they have become a Jeju horse species with a distinct appearance and traits from other external horse species. Our study analyzed genomic structural variation specific to Jeju horses by comparing WGRS data of Jeju horses and Thoroughbred horses. Through the comparative genomic analysis, we have found significant genomic variation in genes that cause differences in disease and behavioral trends in these two different horse species. Overall, there is increasing evidence that endurance genes and immunity genes (i.e., CD1a6 gene) are distinctive traits in Jeju horses.

The most specific discovery in this study is the eqCD1a6 gene. It can be inferred that the eqCD1a6 gene generated strong positive selection in the Jeju horse. Estimating the ratio between synonymous and nonsynonymous mutations is important to understand the dynamics of molecular sequence evolution. Synonymous mutations are unaffected by natural selection in a typical environment, but nonsynonymous mutations are affected by strong natural selection. Given that most genes are strongly influenced by purifying selection, the positive selection of the eqCD1a6 gene was a fascinating phenomenon. As most coding region mutations have a significant effect on the host’s survival, the host typically has a selection process, such as positive selection and purifying selection, for those mutations. As a result, eqCD1a6 in the Jeju horse means that it actively accepts mutations beneficial to survival as some environmental and physiological requirements necessary for survival.

Based on these results, we believe that a positive selection would have occurred in order for the CD1 antigen-binding site of the eqCD1a6 molecule in Jeju horses to recognize specific lipids. The high copy number of eqCD1a homologs involved in immunity to R. equi is very unusual and suggests that CD1a is important to horse immunity. Here, we propose that Jeju horses have evolved to be immunized against indigenous bacteria in Jeju Island [25]. In particular, we confirmed that the region where the most intense positive selection occurred in the eqCD1a6 gene was part of the alpha-chain. Therefore, it is reasonable to assume that positive selection on the eqCD1a6 gene is directed at creating novel receptor–ligand interactions.

## 5. Conclusions

The stimuli to the external environment and avoidance of the stimuli are essential for the survival of animals. That is why positive selection for the more practical side occurs strongly in response to changes in the external environment of living organisms. Compared to widely expressed genes, eqCD1a6 is thought to have fewer restrictions on evolution because it is a specific gene.

The present study identifies region-specific genetic variants in Jeju horses and supports the idea that SNPs present in eqCD1a6 could be used as Jeju horse-specific molecular markers using digital PCR.

## Figures and Tables

**Figure 1 animals-11-01924-f001:**
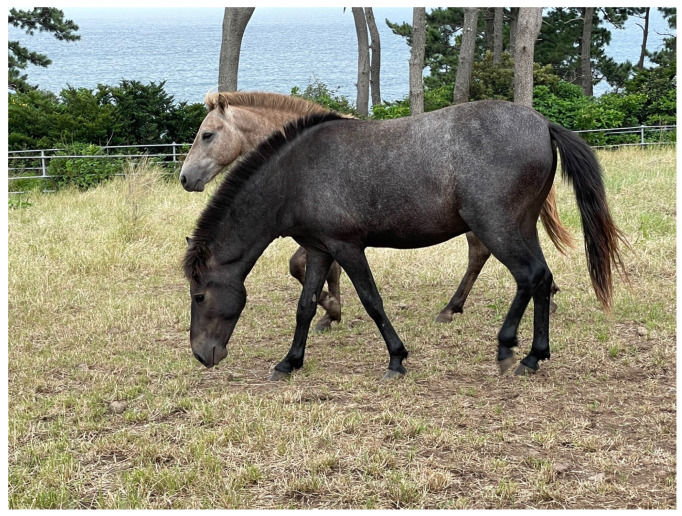
The appearance of pure Jeju horses. This representative photo of a Jeju horse was taken at a grazing ranch on Jeju Island, Korea. As described in the text, the Jeju horse has a small height, a large head, and a thick, short morphology.

**Figure 2 animals-11-01924-f002:**
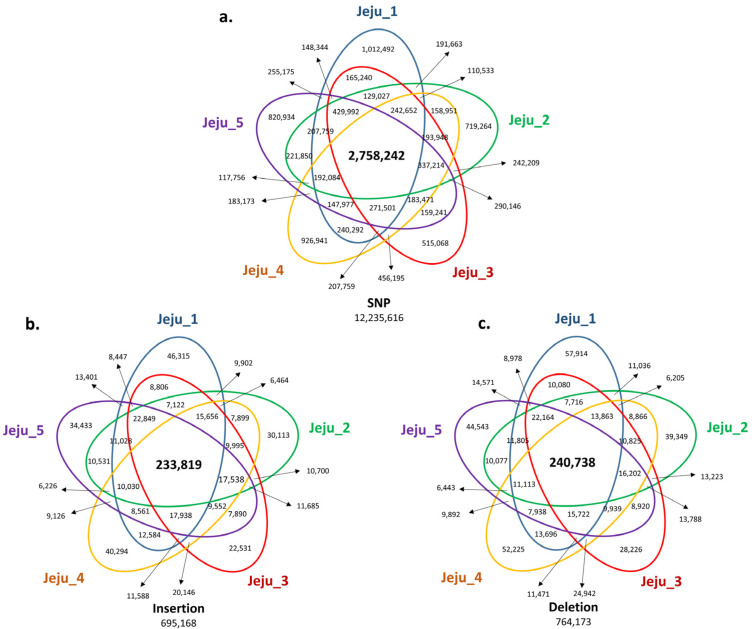
Venn diagram of genomic variants identified from genome comparison with the horse reference genome. In comparative analysis with the horse reference genome (*Equus caballus*, equCab3: January 2018), we screened the number of SNPs and INDELs to find unique variants shared with the five Jeju horses. The number of SNPs (**a**), small insertions (**b**), and deletions (**c**), which are common in the five Jeju horses compared with the horse reference genome, are annotated at each Venn diagram. The number of variants highlighted in the bold letter is common to Jeju horse. (**a**) For SNPs, common variants are 2,758,242 loci, which account for 22.54% of the total. (**b**) For small insertions, 233,819 loci, 33.63% of the total. (**c**) For small deletions, 240,738 loci, 31.5% of the total.

**Figure 3 animals-11-01924-f003:**
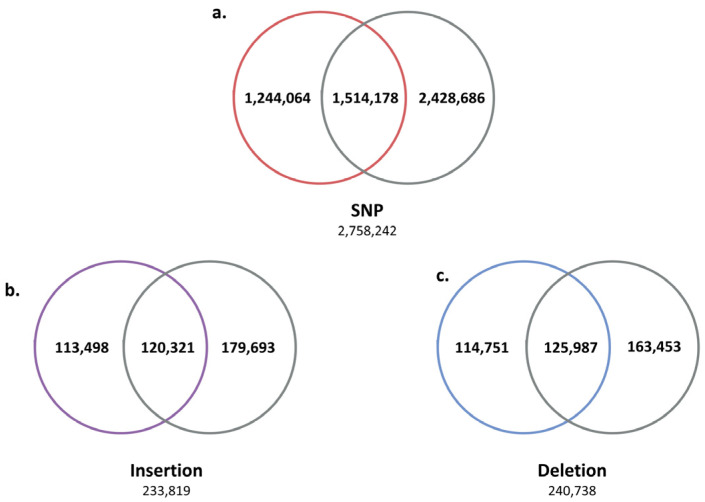
Second comparison of genomic variants between Jeju horse and Thoroughbred. Using the variants compared to the horse reference genes for the first time, secondly, we selected unique variants for the five Jeju horses by comparing and analyzing the genome data of one Thoroughbred obtained in this study. Through this process with another Thoroughbred genome, we were able to accurately distinguish the variant calling of Jeju horse. (**a**) Among the 2,758,242 SNPs identified from the first comparison, a total of 1,244,064 (45.1%) were unique in Jeju horse. (**b**) A total of 113,498 insertions (45.1%) and (**c**) 114,751 deletions (48.54%) were detected in Jeju horse.

**Figure 4 animals-11-01924-f004:**
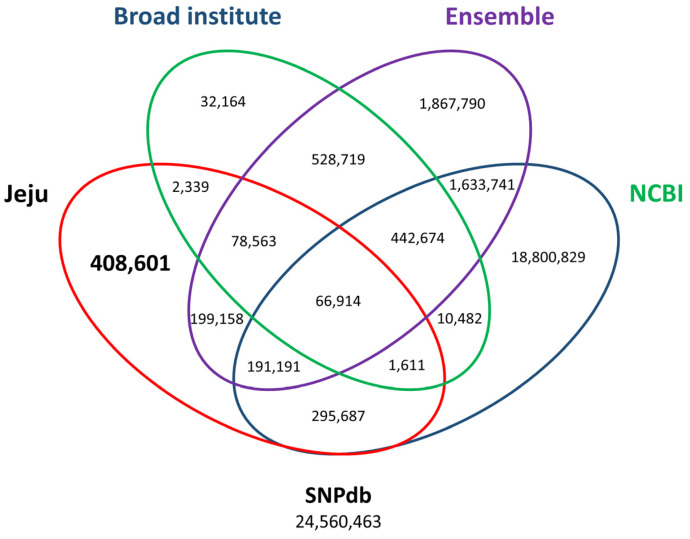
Third comparison of SNP variants using open-access SNP data for horses. By comparing open access SNP data published at the Broad Institute (add URL), Ensembl (add URL), and NCBI (add URL), 408,601 out of 1,244,064 SNPs accounting for 1.66% of all the variants in the SNP database (SNPdb) were finally identified. The number of final Jeju horse-specific SNPs is highlighted in the bold letter.

**Figure 5 animals-11-01924-f005:**
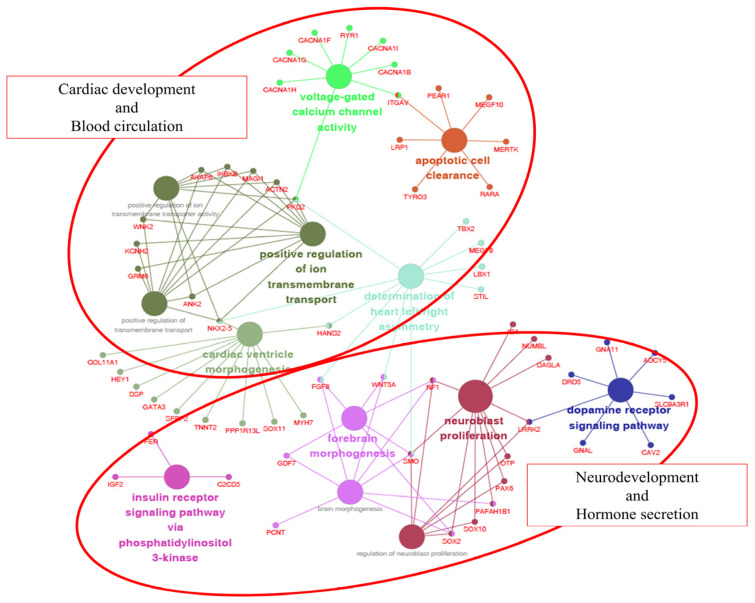
Gene ontology (GO) enrichment analysis of genes with non-synonymous SNPs in Jeju horses. A total of 106 of 788 are significantly associated with cardiac development and blood circulation. Each term is represented by a circle node, where its size is proportional to the number of input genes falling into that term. Its color represents its GO cluster identity (i.e., nodes of the same color belong to the same cluster). The small nodes interacting with circle nodes denote the genes that show associations with the GO cluster.

**Figure 6 animals-11-01924-f006:**
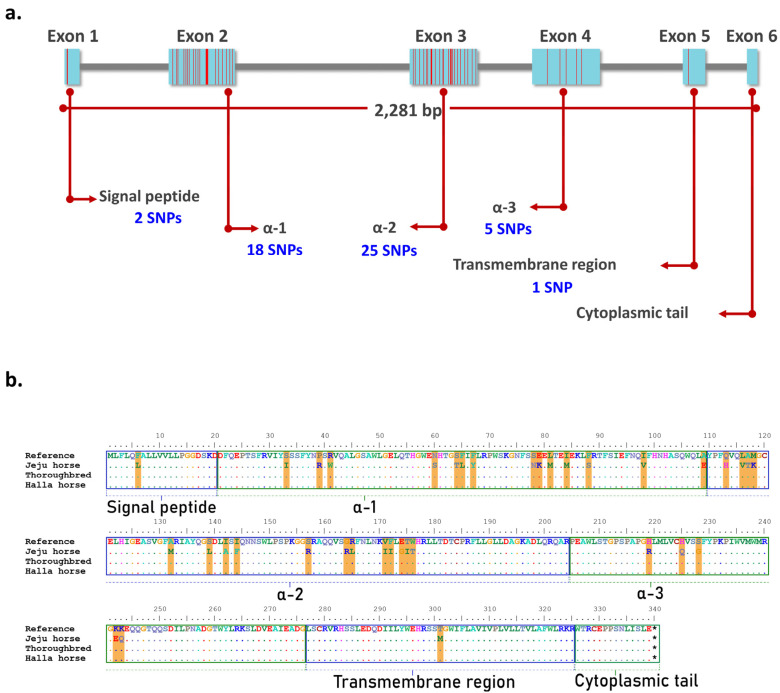
Amino acid (AA) sequence comparison of eqCD1A6 gene. Using BioEdit, eqCD1A6 sequences were visually compared to detect sequence signatures that distinguished between the two species. (**a**) The location of Jeju horse-specific SNPs in the exon region of the eqCD1a6 gene. Most of the SNPs are located in exon 2 and exon 3. (**b**) AA sequences of the eqCD1A6 gene searched by NCBI blast were aligned. Dots indicate AA sequences identical in nine eqCD1A6 isoforms. Orange lines denote the 37 conservative AA substitution spots in Jeju-horses. The eqCD1A6 protein domains, including the signal peptide, α-1, α-2, α-3, transmembrane, and cytoplasmic tail, are boxed with annotations.

**Figure 7 animals-11-01924-f007:**
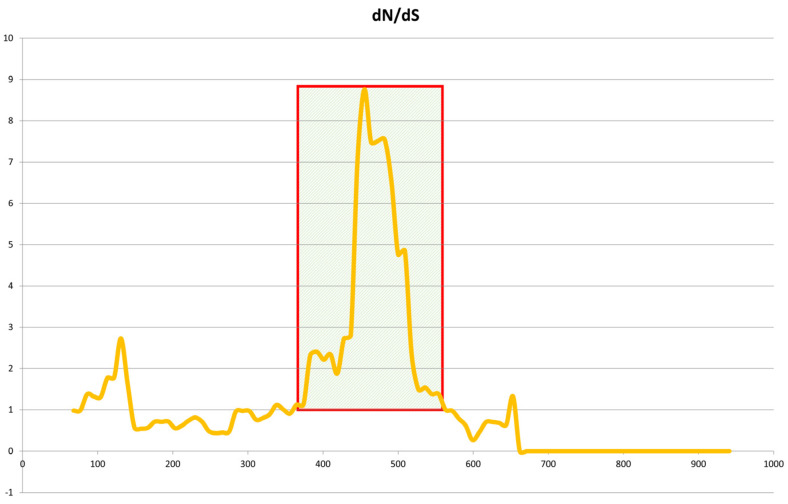
Sliding-window analysis of Jeju horse and Thoroughbred eqCD1a6 genes. Sliding-window analysis of dN/dS ratios was performed along the length of the eqCD1a6 gene, comparing Jeju horse and Thoroughbred gene sequences. dN/dS is plotted against base-pair coordinates in the coding sequence. dN/dS ratios of 1.0 indicate neutral evolution, while ratios of <1.0 are indicative of purifying selection.

**Figure 8 animals-11-01924-f008:**
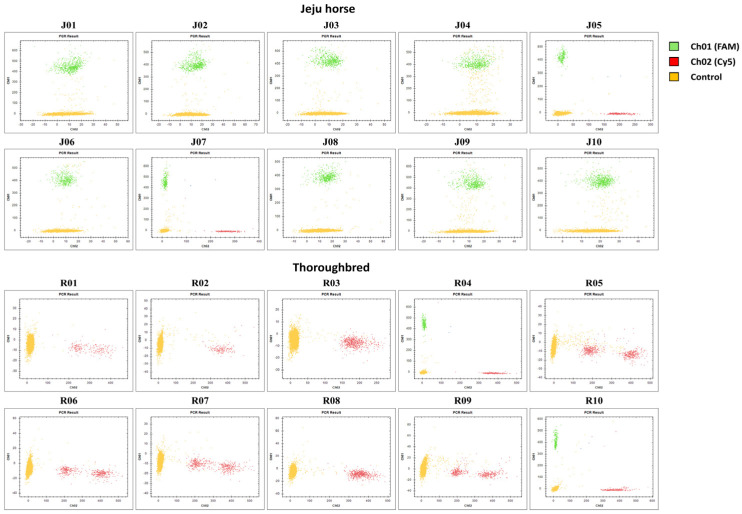
A molecular marker was applied to Jeju horse and Thoroughbred samples in digital PCR assays. A schematic dot plot diagram showing the molecular marker result. The yellow cluster on the plot expresses the control droplets. The green cluster (FAM) and red cluster (SFC620) express the positive droplets for Jeju horse-specific and Thoroughbred-specific, respectively.

**Table 1 animals-11-01924-t001:** The genomic mapping results by whole-genome resequencing.

Classification	Jeju_1	Jeju_2	Jeju_3	Jeju_4	Jeju_5	Thoroughbred	Average
Total reads ^a^	690,508,306	609,871,934	661,791,072	617,836,736	606,111,944	990,419,468	696,089,910
Clean reads	621,253,885	532,985,097	581,568,986	555,881,352	545,197,568	820,451,365	609,556,376
Clean reads, % ^b^	89.97%	87.39%	87.88%	89.97%	89.95%	82.84%	88%
Mapped reads	612,618,456	523,657,858	572,263,882	547,987,837	536,038,249	780,311,183	595,479,578
Mapped reads, % ^c^	98.61%	98.25%	98.40%	98.58%	98.32%	95.11%	97.88%
Average Depth	37.24x	31.48x	34.50x	33.14x	32.36x	46.01x	35.79x

^a^ Total reads: The total number of reads generated. ^b^ The number of reads after trimming and deduplication with Sickle and Picard; (%) = No. of Clean reads/No. of Total reads. ^c^ The number of reads mapped to the reference using BWA mapping tool; (%) = No. of Mapped reads/No. of Clean reads.

**Table 2 animals-11-01924-t002:** The number of variants counting by types for all 5 Jeju horses.

Type ^a^	Jeju_1	Jeju_2	Jeju_3	Jeju_4	Jeju_5	Thoroughbred
Homo INS	246,788	235,661	242,743	238,128	239,981	157,841
Hetero INS	199,722	187,896	193,519	201,288	195,073	142,173
Total INS	446,510	423,557	436,262	439,416	435,054	300,014
Homo DEL	263,896	254,611	259,821	255,380	258,247	166,728
Hetero DEL	201,114	190,505	196,679	204,404	196,586	122,802
Total DEL	465,010	445,116	456,500	459,784	454,833	289,530
Homo SNP	2,404,789	2,331,653	2,415,831	2,329,651	2,369,765	1,740,637
Hetero SNP	4,304,466	4,211,637	4,312,941	4,397,561	4,355,094	2,202,227
Total SNP	6,709,255	6,543,290	6,728,772	6,727,212	6,724,859	3,942,864

^a^ Homo = homozygous; Hetero = heterozygous; INS = insertion; DEL = deletion.

**Table 3 animals-11-01924-t003:** Number of effects by region.

Region	SNP	INS	DEL
Upstream	7715	10,497	7336
Exon	1138	1484	903
Intergenic	63,497	84,897	46,518
intragenic	119	190	112
Intron	21,312	37,272	19,557
Splice site	194	595	435
5’ UTR	174	307	197
3’ UTR	43	106	57
Total	94,192	135,348	75,115

**Table 4 animals-11-01924-t004:** GO functional enrichment analysis of Jeju horse-specific SV.

Gene Ontology	Count	Gene Symbol	*p*-Value	Adjusted *p*-Value
GO:0003208 Cardiac ventricle morphogenesis	11	COL11A1, DSP, GATA3, HAND2, HEY1, MYH7, NKX2-5, PPP1R13L, SFRP2, SOX11, TNNT2	0.000560	0.032501
GO:0005245 Voltage-gated calcium channel activity	8	CACNA1B, CACNA1F, CACNA1G, CACNA1H, CACNA1I, ITGAV, PKD2, RYR1	0.000835	0.046773
GO:0061371 Determination of heart left/right asymmetry	10	FGF8, HAND2, LBX1, MEGF8, NKX2-5, PKD2, SMO, STIL, TBX2, WNT5A	0.000593	0.033776
GO:0007212 Dopamine receptor signaling pathway	7	ADCY5, CAV2, DRD5, GNA11, GNAL, LRRK2, SLC9A3R1	0.000292	0.018073
GO:0038028 Insulin receptor signaling pathway via phosphatidylinositol 3-kinase	3	C2CD5, FER, IGF2	0.000252	0.016145
GO:0043277 Apoptotic cell clearance	7	ITGAV, LRP1, MEGF10, MERTK, PEAR1, RARA, TYRO3	0.000210	0.013660
GO:0048853 Forebrain morphogenesis	6	FGF8, GDF7, NF1, SMO, SOX2, WNT5A	0.000426	0.025571
GO:0048854 Brain morphogenesis	8	FGF8, GDF7, NF1, PAFAH1B1, PCNT, SMO, SOX2, WNT5A	0.000531	0.031340
GO:0007405 Neuroblast proliferation	11	DAGLA, ID4, LRRK2, NF1, NUMBL, OTP, PAFAH1B1, PAX6, SMO, SOX10, SOX2	0.000068	0.004559
GO:1902692 Regulation of neuroblast proliferation	7	LRRK2, NF1, OTP, PAX6, SMO, SOX10, SOX2	0.000900	0.049499
GO:0034764 Positive regulation of transmembrane transport	10	ACTN2, AKAP6, ANK2, GRM6, IKBKB, KCNH2, MAGI1, NKX2-5, PKD2, WNK2	0.000276	0.017401
GO:0034767 Positive regulation of ion transmembrane transport	10	ACTN2, AKAP6, ANK2, GRM6, IKBKB, KCNH2, MAGI1, NKX2-5, PKD2, WNK2	0.000181	0.011963
GO:0032414 Positive regulation of ion transmembrane transporter activity	8	ACTN2, AKAP6, ANK2, IKBKB, MAGI1, NKX2-5, PKD2, WNK2	0.000324	0.019737

**Table 5 animals-11-01924-t005:** Classification of eqCD1a6 SNPs.

Gene	Classification	No. of SNP
	SNPs	275
	Upstream gene variant	162
	Exon variant	51
	Intron variant	62
eqCD1a6	INDELs	21
	Upstream gene variant	19
	Intron variant	2
	Total	296

**Table 6 animals-11-01924-t006:** Prediction of functional effects of the eqCD1a6 gene.

Protein	Position	AA_1_	AA_2_	Description	Prediction Score
eqCD1a6	6 (Exon 1)	F	L	Unknown	-
	33 (Exon 2)	S	I	Benign	0.000
	39 (Exon 2)	P	R	Benign	0.000
	41 (Exon 2)	R	W	Benign	0.000
	60 (Exon 2)	N	S	Benign	0.001
	64 (Exon 2)	S	T	Benign	0.002
	65 (Exon 2)	F	L	Benign	0.128
	67 (Exon 2)	F	Y	Benign	0.156
	78 (Exon 2)	S	N	Benign	0.000
	79 (Exon 2)	E	K	Benign	0.127
	81 (Exon 2)	L	M	Benign	0.311
	84 (Exon 2)	I	M	Benign	0.007
	88 (Exon 2)	F	S	Benign	0.452
	98 (Exon 2)	I	V	Benign	0.000
	109 (Exon 2)	A	E	Benign	0.000
	113 (Exon 3)	Q	H	Possibly damaging	0.871
	116 (Exon 3)	L	V	Benign	0.003
	117 (Exon 3)	A	T	Benign	0.009
	118 (Exon 3)	M	K	Benign	0.001
	132 (Exon 3)	A	M	Benign	0.002
	139 (Exon 3)	S	L	Benign	0.311
	142 (Exon 3)	I	A	Benign	0.401
	144 (Exon 3)	I	F	Benign	0.001
	157 (Exon 3)	S	R	Benign	0.044
	164 (Exon 3)	G	R	Benign	0.000
	165 (Exon 3)	R	L	Benign	0.000
	171 (Exon 3)	V	I	Benign	0.021
	172 (Exon 3)	F	I	Benign	0.001
	174 (Exon 3)	E	G	Possibly damaging	0.950
	175 (Exon 3)	T	I	Benign	0.001
	176 (Exon 3)	W	T	Benign	0.001
	219 (Exon 4)	H	R	Benign	0.001
	225 (Exon 4)	H	Q	Probably damaging	1.000
	228 (Exon 4)	S	G	Benign	0.000
	242 (Exon 4)	K	E	Benign	0.000
	243 (Exon 4)	K	Q	Benign	0.001
	301 (Exon 5)	T	M	Benign	0.016

AA_1_ = reference AA residue; AA_2_ = substitution AA residue.

**Table 7 animals-11-01924-t007:** Quantification of a Jeju horse-specific molecular marker by digital PCR.

Species	Sample	Total Well	Valid Well	Ch01(FAM)			Ch02(Cy5)			Ratio (Ch01/Ch02)	Result
				Positive-Well	Negative-Well	Concentration	Positive-Well	Negative-Well	Concentration		
Jeju horse	J01	20,163	18,781	653	18,128	308.62	0	18,781	0	0	Jeju horse
	J02	20,163	18,430	615	17,815	295.98	0	18,430	0	0	Jeju horse
	J03	20,163	17,464	624	16,840	317.31	0	17,464	0	0	Jeju horse
	J04	20,163	18,389	615	17,774	296.65	0	18,389	0	0	Jeju horse
	J05	20,163	17,790	258	17,532	127.4	255	17,535	125.91	1.01	Heterozygote
	J06	20,163	16,825	414	16,411	217.27	0	16,825	0	0	Jeju horse
	J07	20,163	18,703	256	18,447	120.19	264	18,439	123.98	0.97	Heterozygote
	J08	20,163	17,508	539	16,969	272.7	0	17,508	0	0	Jeju horse
	J09	20,163	18,835	549	18,286	257.97	0	18,835	0	0	Jeju horse
	J10	20,163	19,167	733	18,434	340.06	0	19,167	0	0	Jeju horse
Thoroughbred	R01	20,163	18,413	0	18,413	0	213	18,200	101.47	0	Thoroughbred
	R02	20,163	17,219	0	17,219	0	154	17,065	78.35	0	Thoroughbred
	R03	20,163	18,356	0	18,356	0	588	17,768	283.93	0	Thoroughbred
	R04	20,163	18,365	228	18,137	108.95	256	18,109	122.42	0.89	Heterozygote
	R05	20,163	18,890	0	18,890	0	655	18,235	307.76	0	Thoroughbred
	R06	20,163	17,654	0	17,654	0	468	17,186	234.31	0	Thoroughbred
	R07	20,163	19,244	0	19,244	0	535	18,709	245.88	0	Thoroughbred
	R08	20,163	17,875	0	17,875	0	553	17,322	274.06	0	Thoroughbred
	R09	20,163	18,125	0	18,125	0	486	17,639	237.03	0	Thoroughbred
	R10	20,163	17,613	227	17,386	113.13	260	17,353	129.7	0.87	Heterozygote

## Data Availability

Data supporting reported results can be sent to anyone interested by contacting the corresponding author.

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
