# Peer review of "Comparative Analysis for Genetic Characterization in Korean Native Jeju Horse"

_animals, 2021, doi:10.3390/ani11071924_

Round 1

Reviewer 1 Report

This study aimed to evaluate the genetic diversity of the Jeju horse and preserve its genetic information. To explore Jeju horse-specific genetic differences, the authors made a comparative analysis of large-capacity genomic data between the public database and a thoroughbred variety. They found that the eqCD1a6 gene contains signatures of positive natural selection, which improves immunity to adapt to the barren external environment in Jeju horses. In addition, the authors further validated the Jeju horse-specific SNPs of high accuracy in the aqCD1a6 gene by employing the digital PCR method. There is no problem with the method, but I have some questions about some results.

  1. Line 305-306, the authors mentioned that the accuracy was 80% and it means 20% was wrong. In Figure 7 and Table 7, J05, J07, R04, and R10 were not classified correctly. What are the possible reasons?
  2. If you predict that designing Jeju horse-specific probes based on sequence information of various SNPs was more accurate than single SNP, is there a combination of SNPs to provide the most accurate classification?
  3. In result 3.1, the correct number should be 6,686,678,181 instead of 6,6686,678 181.

Reviewer 2 Report

Dear Authors,

attached please find a pdf with comments.

Regards

Round 2

Reviewer 1 Report

Thank you for revising the manuscript, I have no additional comments.

Reviewer 2 Report

Dear Authors,

accept in present form.

Regards